# Replicated chromatin curtails 53BP1 recruitment in BRCA1-proficient and BRCA1-deficient cells

Jone Michelena[1],*, Stefania Pellegrino[1,2],*, Vincent Spegg[1,2],*, Matthias Altmeyer[1]

DNA double-strand breaks can be repaired by non-homologous end-joining or homologous recombination. Which pathway is used depends on the balance between the tumor suppressors 53BP1 and BRCA1 and on the availability of an undamaged template DNA for homology-directed repair. How cells switch from a 53BP1-dominated to a BRCA1-governed homologous re-combination response as they progress through the cell cycle is incompletely understood. Here we reveal, using high-throughput microscopy and applying single cell normalization to control for increased genome size as cells replicate their DNA, that 53BP1 recruitment to damaged replicated chromatin is inefficient in both BRCA1-proficient and BRCA1-deficient cells. Our results substantiate a dual switch model from a 53BP1-dominated response in unreplicated chromatin to a BRCA1–BARD1–dominated response in replicated chromatin, in which replication-coupled dilution of 53BP1's binding mark H4K20me2 functionally cooperates with BRCA1–BARD1–mediated suppression of 53BP1 binding. More generally, we suggest that appropriate normalization of single cell data, for example, to DNA content, provides additional layers of information, which can be critical for quantifying and interpreting cellular phenotypes.

## Introduction

The balance between non-homologous end-joining (NHEJ) and homologous recombination (HR) has important implications for maintenance of genome stability, for exploiting DNA damage re-sponse (DDR) defects as vulnerabilities in cancer therapy, and for harnessing the full potential of CRISPR/Cas9–mediated genome editing and gene therapy (Panier & Boulton, 2014; O'Connor, 2015; Fellmann et al, 2017). The choice between NHEJ and HR is closely linked to the cell cycle and to whether or not an undamaged homologous stretch of DNA is available as a template for HR (Hustedt & Durocher, 2017; Her & Bunting, 2018; Murray & Carr, 2018). While 53BP1–RIF1–Shieldin restrains DNA end resection of DNA

double-strand breaks (DSBs) in the absence of a homologous template strand, after DNA replication BRCA1–BARD1 counteracts 53BP1 binding to damaged chromatin and promotes HR reactions (Chapman et al, 2012a; Ochs et al, 2016; Pellegrino et al, 2017; Simonetta et al, 2018; Nakamura et al, 2019; Setiaputra & Durocher, 2019). The antagonism between 53BP1–RIF1–Shieldin and BRCA1–BARD1 has important implications for cancer therapy, in particular for targeting BRCA–deficient tumors with poly(ADP-ribose) poly-merase (PARP) inhibitors, and for improving CRISPR/Cas9–mediated gene editing.

HR is confined to the S and G2 phase of the cell cycle, yet how the DDR discriminates between replicated and unreplicated areas of the genome during S phase progression has only started to emerge recently. Such discrimination is important for HR reactions to be favored when DSBs occur in replicated DNA, and mutagenic end resection to be prevented when DSBs occur in areas of the genome that were not replicated yet. Accumulating evidence suggests that this discrimination is linked to the different chromatin makeup ahead of and behind replication forks (Alabert et al, 2014; Saredi et al, 2016; Pellegrino et al, 2017; Simonetta et al, 2018; Nakamura et al, 2019). Of particular interest for the antagonism between the 53BP1 and BRCA1 protein complexes is the H4K20 dimethylation mark (H4K20me2), which is abundant in unreplicated chromatin on parental histones and absent from newly incorporated histones. Hence, H4K20me2 is diluted in nascent replicated chromatin and only restored in mature chromatin in late G2/M (Alabert et al, 2014; Saredi et al, 2016; Pellegrino et al, 2017; Simonetta et al, 2018; Nakamura et al, 2019). 53BP1 binds H4K20me2 via its tandem Tudor domain, and both H4K20me2 and the tandem Tudor domain are required for 53BP1 recruitment to DSBs (Lukas et al, 2011; Pellegrino & Altmeyer, 2016; Schwertman et al, 2016). Conversely, unmethylated H4K20 (H4K20me0) in replicated chromatin is recognized by the HR-promoting protein complexes TONSL–MMS22L and BRCA1–BARD1 (Saredi et al, 2016; Nakamura et al, 2019).

Among the evolving methods to probe cellular stress re-sponses at the single cell level is quantitative image-based cytometry (QIBC), which uses automated high-content microscopy in a cytometry-like fashion to stage individual cells according to their position in the cell cycle (Toledo et al, 2013; Michelena et al, 2018;

---

[1]Department of Molecular Mechanisms of Disease, University of Zurich, Zurich, Switzerland   [2]Life Science Zurich Graduate School (LSZGS), Zurich, Switzerland

Correspondence: matthias.altmeyer@uzh.ch
*Jone Michelena, Stefania Pellegrino, and Vincent Spegg contributed equally to this work

Teloni et al, 2019). Besides quantification of multiple cellular parameters in large-cell populations, when such microscopy-based single-cell measurements are accurate enough to discriminate cells according to their position in the cell cycle, they allow for cell cycle–related data normalization, for example to the size of a cell or its nucleus, or to the replication status of the genome. As multiple cellular functions are tightly linked to cell cycle progression and to the associated duplication of cellular contents, we suggest, using

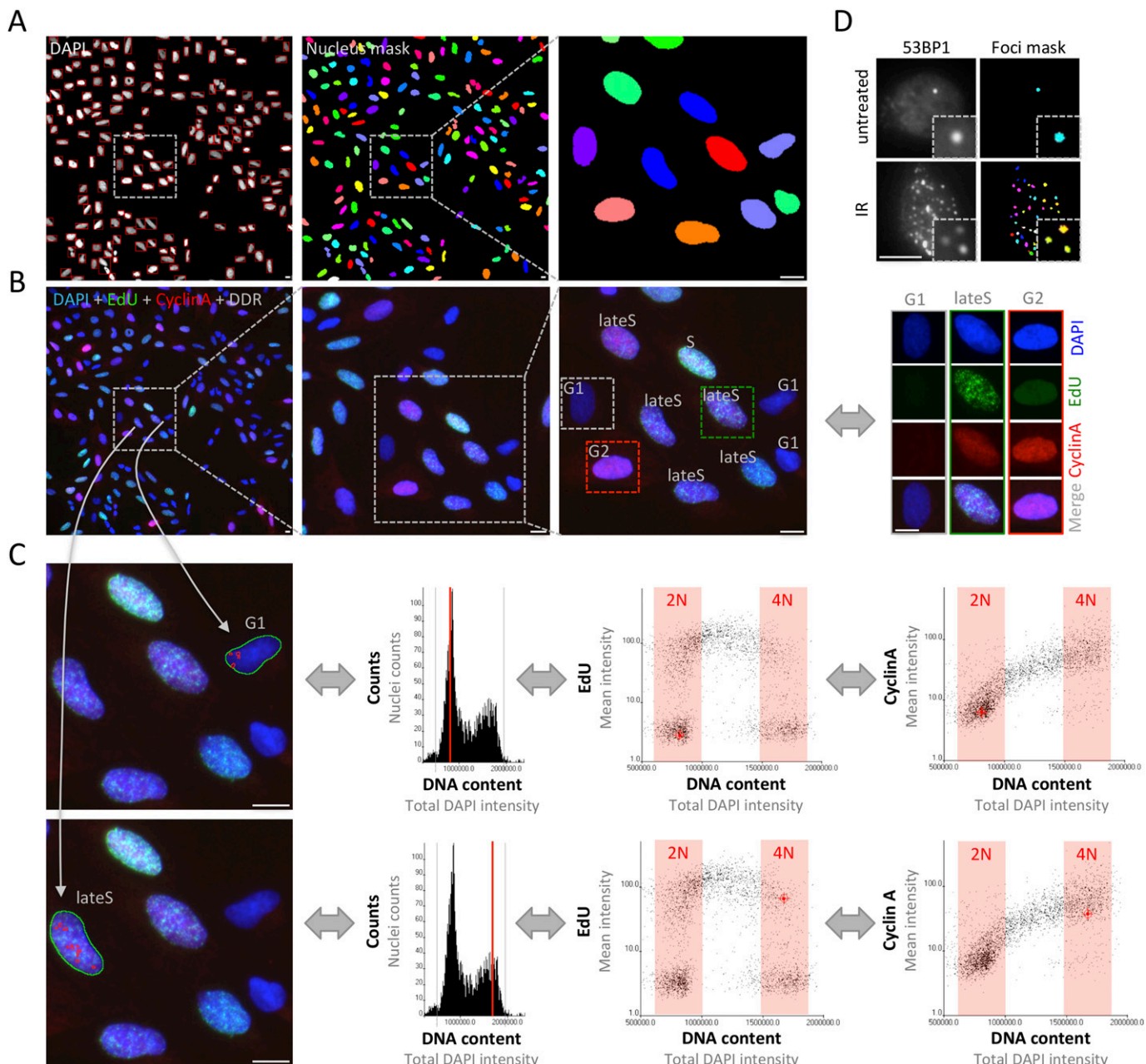

**Figure 1.   Cell cycle staging by quantitative image-based cytometry.**
**(A)** Image segmentation based on the DAPI signal to detect individual cell nuclei. Typically, between 50 and 100 images per condition are acquired, yielding image information of between 3,000 and 10,000 cells. **(B)** In multi-color imaging experiments, appropriate cell cycle markers (here DAPI, EdU, and Cyclin A) are combined with markers of interests to allow cell cycle–resolved interrogations of cellular responses. When multiple cell cycle markers are rationally combined and cross-compared, precise cell cycle staging can be achieved (e.g., to discriminate early G2 from late G2 cells or late G2 cell from mitotic cells). **(C)** Cell cycle profiles are generated for each cell population (here a one-dimensional cell cycle profile based on DAPI as well as two-dimensional cell cycle profiles based on EdU versus DAPI and Cyclin A versus DAPI). In the upper micrograph, a single cell in G1 is selected and its cell cycle position in the cell cycle profiles to the right is indicated in red. In the lower micrograph, a cell in late S-phase is selected and its cell cycle position in the cell cycle profiles to the right is indicated in red. As images and numerical data are linked, the analysis works in both ways, from single cell images to cell cycle profiles, and reciprocally from cell cycle profiles to single cell images, allowing for both quantification-based and image-based explorations of cell cycle–related phenotypes. **(D)** For each individual cell nucleus, sub-nuclear structures such as ionizing radiation-induced foci (here 53BP1) are segmented and their number and signal intensities are quantified to yield cell cycle–resolved maps of DNA damage responses. Scale bars, 10 μm.

the example of cell cycle–regulated 53BP1 recruitment to damaged chromatin as a paradigm, that normalization of cell cycle-related phenotypes to DNA content and thus cell cycle position can reveal additional layers of information that facilitate biological interpretation of quantitative cell imaging data.

It was recently established that the ankyrin repeat domain of BARD1 binds to H4K20me0 and brings the BRCA1–BARD1 complex to DNA breaks, thus illuminating the mechanistic underpinnings of BRCA1-mediated displacement of 53BP1 from replicated chromatin (Nakamura et al, 2019). An unresolved question is, however, whether the antagonism between BRCA1–BARD1 and 53BP1 alone is sufficient to explain the displacement of 53BP1 from damaged replicated chromatin, or whether the dilution of H4K20me2 behind replication forks directly impacts 53BP1 recruitment. Elucidating the relative contribution of BRCA1–BARD1–related versus unrelated factors for antagonizing 53BP1's functions could provide critical information on DSB repair pathway choice with implications for using HR and NHEJ in the context of genome editing and for targeting the DDR in cancer therapy.

## Results

By means of software-assisted image segmentation and feature extraction, QIBC allows measurements of cellular phenotypes in large, asynchronously growing cell populations. Segmentation masks defined by the DAPI signal can be used to detect individual cell nuclei (Fig 1A). By multicolor imaging, additional cell cycle markers, such as the DNA replication marker 5-Ethynyl-2′-deoxyuridine (EdU) and Cyclin A (other useful markers include Cyclin B and the mitotic marker H3pS10), can be acquired in conjunction with a DDR marker of interest (Fig 1B). Validation of correct cell cycle staging at the level of individual cells occurs in both directions, from single cell images to the position of the cell in image-based cell cycle profiles, and conversely from any position in the one- or

two-dimensional cell cycle profiles to the individual cell images (Fig 1C). Additional segmentation masks can be applied to detect sub-nuclear structures such as ionizing radiation (IR) induced foci (Fig 1D). By use of appropriate markers, phenotypes at any particular position in the cell cycle can thus be interrogated, without the need for cell cycle perturbations such as synchronization and without the need to categorize (i.e., gate) cells into binned groups. Of note, in such experiments, the total DAPI intensity per nucleus scales linearly with DNA content, doubling as cells go from G1 (2N) to G2 (4N) and can therefore be used as a direct measure of genome size (Fig 1C).

Scoring IR-induced 53BP1 foci by QIBC, we previously reported a gradual decline in 53BP1 accumulation at damaged chromatin as cells go through the S-phase (Pellegrino et al, 2017), consistent with other studies (Chapman et al, 2012a; Simonetta et al, 2018). IR-induced DNA damage scales with the amount of DNA present in a cell's nucleus and doubles as cells replicate their genome (Fig 2A). Consistently, when we measured DNA content and γH2AX foci formation (15 min after IR) as marker of IR-induced DSBs in a cell cycle–resolved manner, both DNA content and γH2AX foci approximately doubled as cells went from G1 to G2 (Fig 2B and C). This is in agreement with prior work on irradiation-induced DNA damage load scaling linearly with the amount of DNA exposed to IR (Rothkamm et al, 2003; Kruger et al, 2004; Wardman et al, 2007; Bauerschmidt et al, 2010; Barnard et al, 2013), and with standard normalization procedures when working with cell populations synchronized at different phases of the cell cycle (e.g., for DSB measurements by pulse field gel electrophoresis twice as many G1-arrested cells as G2-arrested cells are loaded on the gel to correct for the difference in DNA content [Kruger et al, 2004]).

We thus decided to take a closer look at the recruitment of 53BP1 in presence or absence of its antagonist BRCA1–BARD1, taking into account that the induced DNA damage is proportional to the amount of DNA present in the nucleus. As expected, single-cell normalization to DNA content in BRCA1–BARD1–proficient cells revealed a pronounced decline in 53BP1 foci formation as cells go

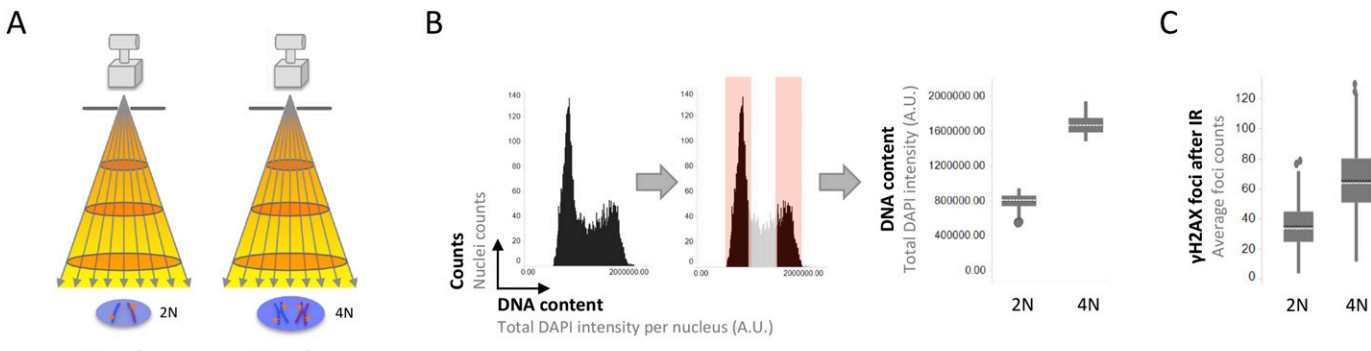

**Figure 2. Ionizing radiation (IR)-induced DNA damage scales with DNA content.**
**(A)** Scheme to illustrate that IR induced DNA damage scales with genome size. Because of the increased genome size when comparing G2 cells with G1 cells, more DNA damage occurs in an irradiated G2 cell nucleus compared with a G1 cell nucleus. **(B)** Quantitative image-based cytometry allows for cell cycle profiling based on the DAPI signal as proxy of DNA content. Accordingly, the DAPI signal in U-2 OS cells with a 4N DNA content (G2) is twice as high as the DAPI signal in cells with a 2N DNA content (G1). **(C)** Consistently, approximately twice as many γH2AX foci, as marker of DNA damage, are quantified in G2 cells as compared with G1 cells at early time-points after IR. U-2 OS cells were treated with 0.5 Gy of IR, fixed 15 min later, stained for DNA content and γH2AX, and γH2AX foci were quantified in a cell cycle–resolved manner by quantitative image-based cytometry. Box plots with medians and averages are shown.

through S-phase (Figs S1A–D and S2A–D). Nuclear 53BP1 levels were stable across the cell cycle (Fig S2E). Consistent with previous work (Bromberg et al, 2017; Pellegrino et al, 2017), the SUV4-20 inhibitor A-196, which as a competitive inhibitor of the histone methyltransferases SUV4-20H1/2 blocks re-establishment of H4K20me2/3 in post-replicative chromatin, reduced IR-induced 53BP1 foci formation (Fig S3A and B). This effect was most pronounced in replicated chromatin, in agreement with a failure to restore H4K20 methylation after replication-coupled dilution of H4K20me2/3. Reduced 53BP1 foci formation in replicated chromatin was also seen in CRISPR/Cas9–engineered cells expressing fluorescently labeled 53BP1 from its natural gene promoter (Kilic et al, 2019), and the

suppression of 53BP1 recruitment was again even clearer when single cell normalization to DNA content was applied (Fig 3A–E). Importantly, single-cell normalization to DNA content also revealed that IR-induced 53BP1 foci formation gradually declined as a function of DNA replication upon depletion of either BRCA1 or BARD1, although to a lesser extent as in BRCA1–BARD1–proficient cells (Figs 4A–C, S4A–D, and S5A and B). The difference in efficiency of 53BP1 recruitment in gated cell subpopulation averages based on DNA content (2N versus 4N) was even more pronounced when G1 cells were compared with cells in late S/early G2 based on DAPI and Cyclin A staining, both in BRCA1–BARD1–proficient and BRCA1–BARD1–deficient cells (Fig 5A–D). Similarly, when H4K20me2 was

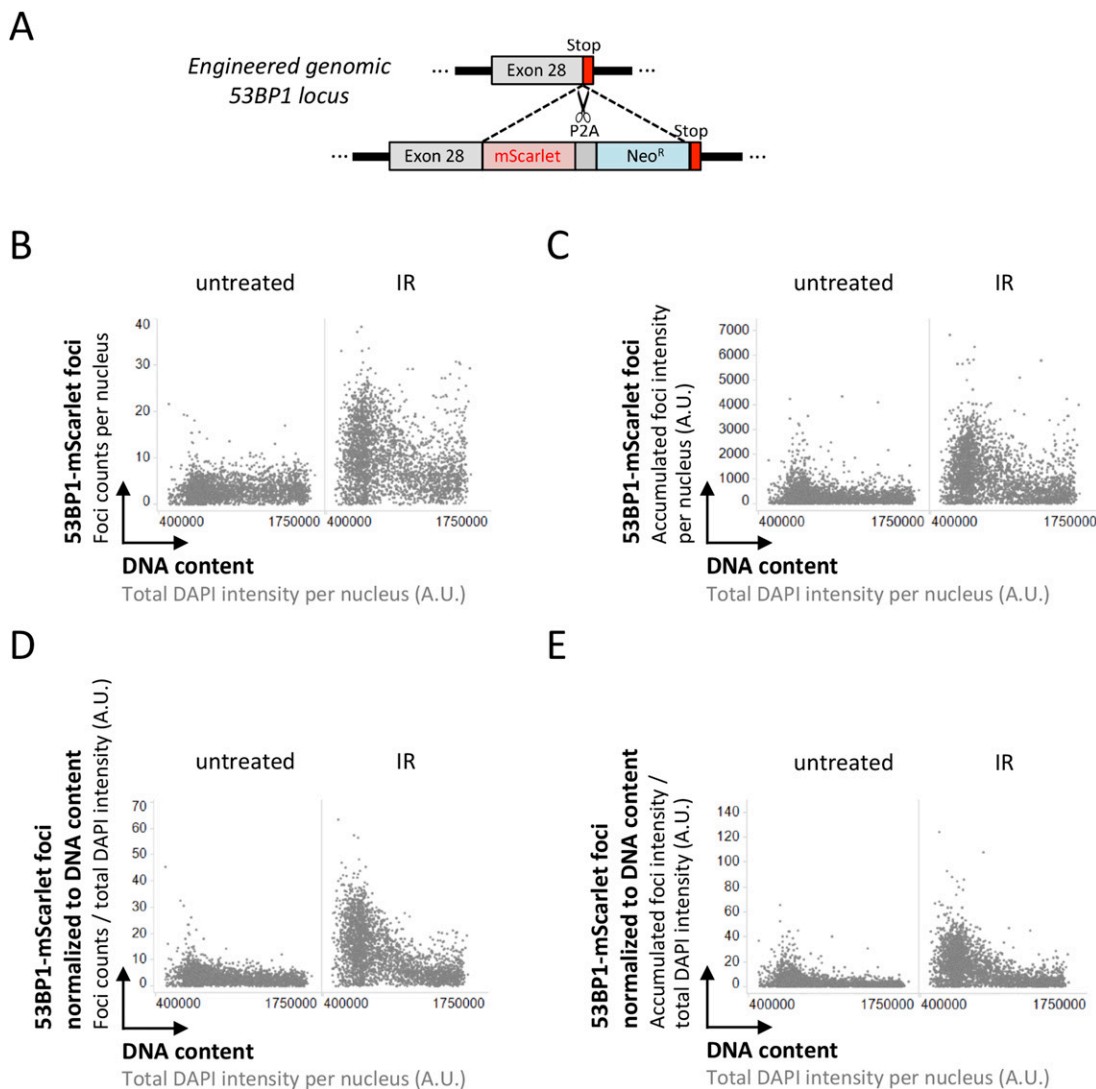

**Figure 3.   53BP1 response to ionizing radiation-induced DNA damage as a function of cell cycle progression.**
**(A)** Scheme of the CRISPR/Cas9-based targeting of the endogenous 53BP1 gene locus to introduce mScarlet and generate cell lines expressing fluorescently labeled 53BP1 from its natural promoter (Kilic et al, 2019). **(B)** U-2 OS 53BP1-mScarlet cells were treated with 0.5 Gy of ionizing radiation, fixed 45 min later, stained for DNA content, and 53BP1-mScarlet foci counts were quantified by quantitative image-based cytometry. **(C)** For the same cells shown in (B) the accumulated intensity of 53BP1-mScarlet foci per nucleus is shown. **(D)** The quantification from (B) was normalized at the single cell level to DNA content to control for increasing damage load with increasing DNA amount (see Fig 2). The parameter resulting from this normalization has arbitrary units and was multiplied by a multiple of 10 to yield data that could be plotted on linear scale in the depicted range. **(E)** For the same cells shown in (B), the accumulated intensity of 53BP1-mScarlet foci per nucleus was normalized to the DNA content of the same nucleus.

used to define G1 versus late S/early G2 cells (i.e., excluding cells in mid and late G2 with gradually restored H4K20me2), the difference in 53BP1 recruitment was more pronounced compared with cells gated merely based on their DNA content (Fig 6A–D). Consistent results were obtained with the breast cancer cell line SUM149PT carrying the *BRCA1* 2288delT mutation and allelic *BRCA1* loss (Fig S6A–C).

The SUV4-20 inhibitor A-196 reduced H4K20me2 similarly in BRCA1-proficient and BRCA1-deficient cells (Fig S7A). As expected for a replication-coupled dilution effect, H4K20me2 was not completely abolished by A-196 treatment and residual H4K20me2 remained, in particular in G1 cells (Fig S7A). Consistent with impaired re-establishment of H4K20me2 after replication, an increase in H4K20me1 was observed in both BRCA1-proficient and BRCA1-deficient cells (Fig S7B). Efficient BRCA1 depletion was confirmed by reduced BRCA1 intensities (Fig S7C) and by abolished BRCA1 foci formation after knockdown (Fig S7D). Normalized 53BP1 recruitment was reduced in cells with a 4N DNA content compared with cells with a 2N DNA content, again both in BRCA1-proficient and BRCA1-deficient cells, and impaired H4K20me2 restoration upon A-196 treatment was associated with reduced 53BP1 recruitment (Fig S7E). 53BP1 recruitment was not completely abolished by A-196 treatment, however, suggesting that the residual H4K20me2 still supported 53BP1 binding and/or that additional methylations might cooperate with H4K20me2 to facilitate 53BP1 chromatin binding via its tandem Tudor domain.

Taken together, we conclude that inefficient 53BP1 recruitment to damaged replicated chromatin is an inherent feature that occurs both in BRCA1-deficient and, in a more pronounced manner, in BRCA1-proficient cells. We therefore suggest that replication-coupled dilution of H4K20me2, in addition to enabling H4K20me0-mediated BRCA1–BARD1 recruitment, also directly affects the efficiency of 53BP1 recruitment in response to DNA damage (Fig 7A–C).

## Discussion

53BP1 requires its oligomerization domain for recruitment to sites of DNA damage and shows features of dynamic self-assembly associated with phase separation (Kilic et al, 2019; Pessina et al, 2019; Piccinno et al, 2019). A reduced concentration of H4K20me2, as present in replicated nascent chromatin likely increases the threshold for efficient 53BP1 accumulation. Conversely, the high concentration of H4K20me2 in unreplicated chromatin (i.e., in G1/G0 cells and in unreplicated regions of the genome during the S-phase progression), together with DNA damage–induced chromatin modifications that promote multivalent 53BP1 chromatin binding, provides a scaffold for efficient 53BP1 assembly around DNA break sites in the absence of a replicated template DNA required for HR repair. We, therefore, suggest that the effect of H4K20me2 dilution (and potentially of additional chromatin features relevant for 53BP1 binding) in nascent replicated chromatin functionally cooperates with the effect of H4K20me0-mediated BRCA1–BARD1 recruitment and the ensuing 53BP1 displacement. Together, they represent a dual switch to ensure that DSBs in unreplicated areas of the genome are protected from excessive DNA end resection and

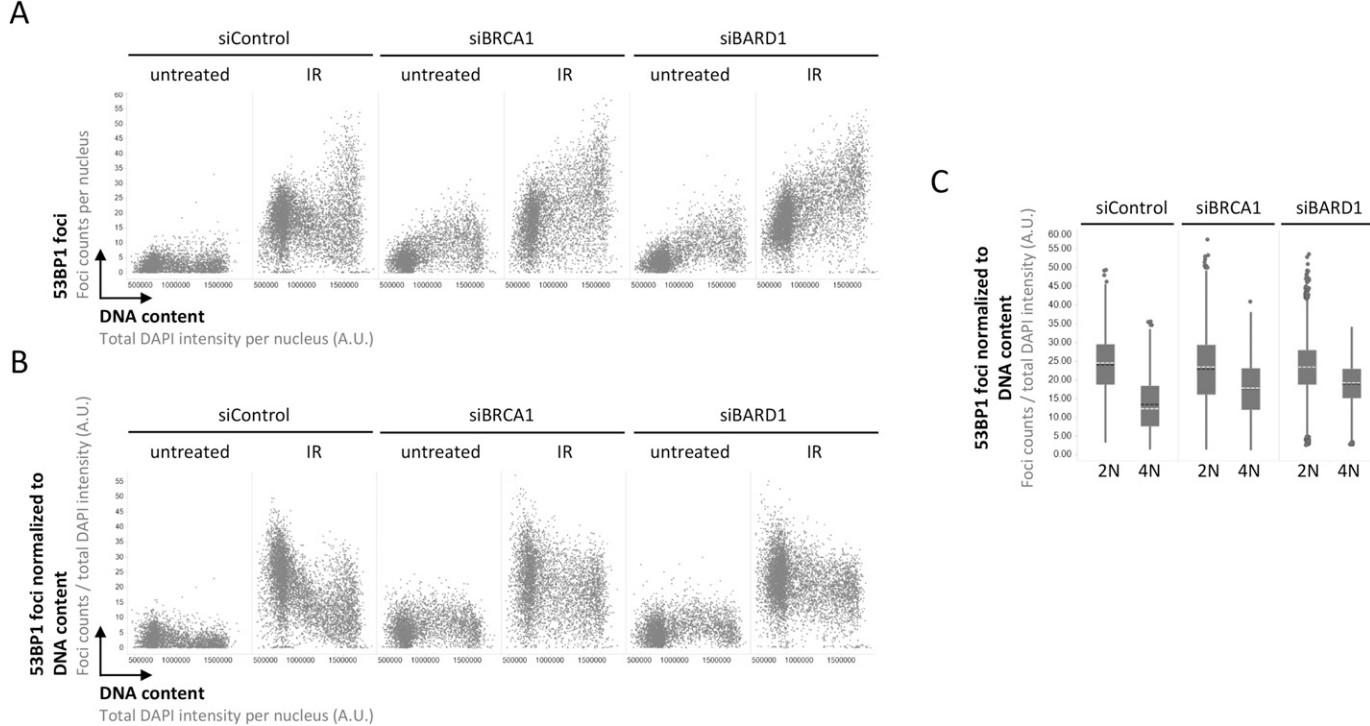

**Figure 4. 53BP1 recruitment to replicated chromatin is inefficient in absence of BRCA1–BARD1.**
**(A)** Cell cycle–resolved analysis of ionizing radiation (IR)-induced 53BP1 foci formation in control conditions and upon depletion of either BRCA1 or BARD1. U-2 OS cells were treated with 0.5 Gy of IR, fixed 45 min later, stained for DNA content and 53BP1, and 53BP1 foci were quantified by quantitative image-based cytometry. **(B)** The quantification from (A) was normalized at the single cell level to DNA content to control for increasing damage load with increasing DNA amount (see Fig 2). **(C)** Averaged relative 53BP1 recruitment after IR from (B) is compared in binned 2N versus 4N cells in presence or absence of BRCA1–BARD1. Box plots with medians and averages are shown.

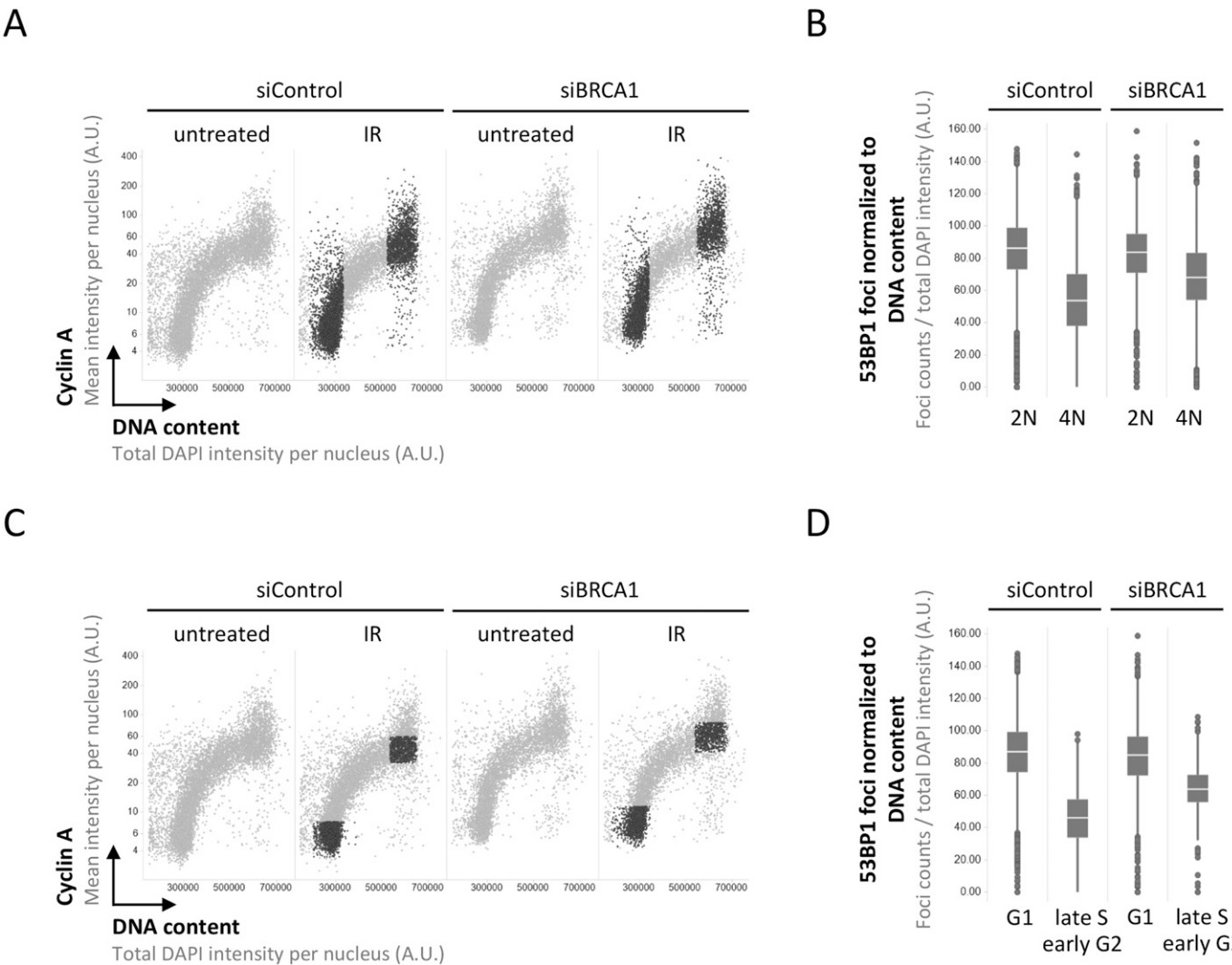

**Figure 5. Quantitative image-based cytometry–assisted cell cycle gating based on DAPI and Cyclin A substantiates impaired 53BP1 recruitment to damaged replicated chromatin in both BRCA1-proficient and -deficient cells.**
**(A)** 2-D cell cycle profiles based on DAPI and Cyclin A. Cells with a 2N and 4N DNA content in the ionizing radiation (IR)-treated samples marked in black. **(B)** Averaged relative 53BP1 recruitment after IR from the cells marked in black in (A) is compared. Box plots with medians are shown. **(C)** 2-D cell cycle profiles based on DAPI and Cyclin A. Cells in G1 and cells in late S/early G2 in the IR-treated samples marked in black. **(D)** Averaged relative 53BP1 recruitment after IR from the cells marked in black in (C) is compared. Box plots with medians are shown.

illegitimate recombination (Mirman & de Lange, 2020) and channeled towards NHEJ, whereas DSBs in replicated areas of the genome are released from the DNA end protection functions of 53BP1 and channeled towards resection and HR (Fig 7A–C).

The function of 53BP1-Shieldin to restrain DNA end resection in unreplicated chromatin may promote both classical NHEJ and, for example, at "dirty" DSBs with ssDNA overhangs or if classical NHEJ fails, also alternative NHEJ (Bothmer et al, 2010; Xiong et al, 2015; Han et al, 2017; Rother et al, 2020). The antagonism between the 53BP1 complex and BRCA1–BARD1 (Callen et al, 2013; Chapman et al, 2013; Di Virgilio et al, 2013; Escribano-Diaz et al, 2013; Feng et al, 2013; Zimmermann et al, 2013) could be seen as a bistable system, the robustness of which is achieved by cooperative effects resulting from high affinity 53BP1 and low affinity BRCA1–BARD1 binding to unreplicated chromatin versus low affinity 53BP1 and high affinity

BRCA1–BARD1 binding in replicated chromatin. This view is consistent with supraphysiological 53BP1 accumulation at damaged replicated chromatin in BRCA1-deficient cells, yet it suggests that even in absence of BRCA1–BARD1 the accumulation of 53BP1 is curtailed by reduced H4K20me2 in replicated chromatin. The DDR, thus, makes use of replication-coupled dilution of an abundant histone mark, which cells only restore in replicated chromatin when genome duplication has been completed. Enforced premature restoration of H4K20me2 during S-phase progression indeed shifts the balance towards a 53BP1-governed response (Pellegrino et al, 2017). The restoration of H4K20me2 in late G2 associated with regained 53BP1 binding is consistent with 53BP1-dependent formation of radial chromosomes in the absence of BRCA1 (Bouwman et al, 2010; Bunting et al, 2010). During S-phase progression, the chromatin-embedded dual switch coming from high affinity 53BP1 versus low affinity

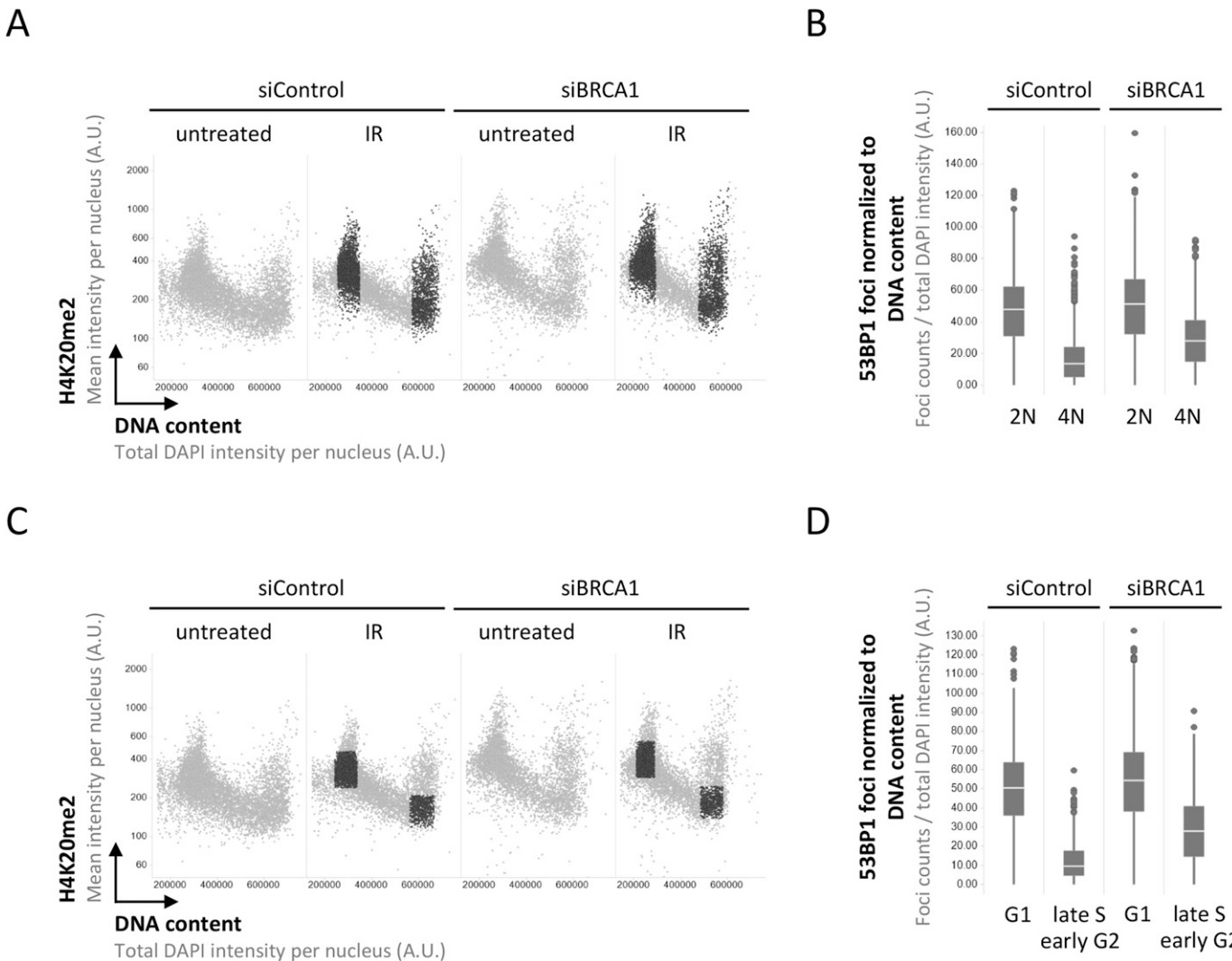

**Figure 6. Quantitative image-based cytometry–assisted cell cycle gating based on DAPI and H4K20me2 substantiates impaired 53BP1 recruitment to damaged replicated chromatin in both BRCA1-proficient and -deficient cells.**
**(A)** 2-D cell cycle profiles based on DAPI and H4K20me2. Cells with a 2N and 4N DNA content in the ionizing radiation (IR)-treated samples marked in black. **(B)** Averaged relative 53BP1 recruitment after IR from the cells marked in black in (A) is compared. Box plots with medians are shown. **(C)** 2-D cell cycle profiles based on DAPI and H4K20me2. Cells in G1 and cells in late S/early G2 in the IR-treated samples marked in black. **(D)** Averaged relative 53BP1 recruitment after IR from the cells marked in black in (C) is compared. Box plots with medians are shown.

BRCA1–BARD1 binding to unreplicated chromatin, which is reverted in replicated chromatin, is likely reinforced by global cell cycle-dependent protein modifications (Chapman et al, 2012b; Tang et al, 2013; Daley & Sung, 2014; Jacquet et al, 2016; Hustedt & Durocher, 2017; Isono et al, 2017; Li et al, 2019; Walser et al, 2020).

In light of the growing interest to target the balance between HR and NHEJ in cancer therapy and to modify the underlying mechanisms for improved genome editing, we envision that the dual switch mechanism from an H4K20me2–53BP1–dominated response in unreplicated chromatin to an H4K20me0–BRCA1–BARD1–dominated response in replicated nascent chromatin will be of relevance.

More generally, we propose that, depending on the biological question, appropriate normalization becomes inevitable to interpret high-content single cell data, and that image-based normalization to cell size, nuclear volume, DNA content, or other suitable cell cycle markers can provide additional layers of information, which may be critical for quantifying and interpreting cellular responses to stress, including genotoxic stress by irradiation, chemotherapy, or newly emerging anticancer drugs.

# Materials and Methods

### Cell culture and treatments

Human U-2 OS cells and U-2 OS cells expressing 53BP1-mScarlet from the endogenous promoter (Kilic et al, 2019) were grown under standard cell culture conditions (humidified atmosphere, 5% $CO_2$) in DMEM containing 10% fetal bovine serum (GIBCO) and 1%

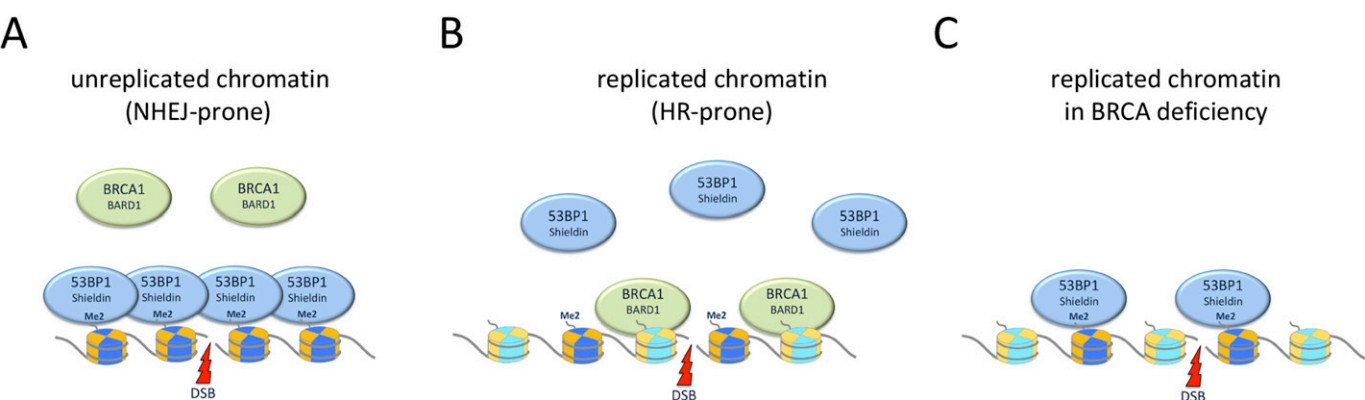

**Figure 7. Simplified model of a dual switch to regulate the accumulation of 53BP1 and BRCA1–BARD1 at unreplicated versus replicated damaged chromatin.**
**(A)** The high density of H4K20me2 in unreplicated chromatin (in G1/G0, and throughout S-phase in yet to be replicated regions of the genome), together with DNA damage induced γH2AX and H2AK15ub (not shown), promotes efficient assembly of 53BP1 and its downstream effectors. This shields double-strand breaks against excessive DNA end resection and illegitimate recombination and generally renders break sites non-homologous end-joining-prone. **(B)** As a dual switch, reduced 53BP1 binding upon replication-coupled dilution of H4K20me2 functionally cooperates with H4K20me0-mediated BRCA1–BARD1 recruitment to promote DNA end resection and homologous recombination reactions in replicated areas of the genome (during S-phase progression in nascent chromatin and prior to H4K20me2 restoration in late G2/M). **(C)** In BRCA1–BARD1–deficient cells, 53BP1 can accumulate at damaged replicated chromatin and exert some of its functions, however, compared to unreplicated chromatin this recruitment is less efficient, likely reflecting the reduced density of H4K20me2.

penicillin–streptomycin antibiotics. SUM149PT cells (kindly provided by Alessandro Sartori) were grown in Ham's F-12 medium (Thermo Fisher Scientific) containing 10% fetal bovine serum and 1% of penicillin–streptomycin antibiotics. All cells were maintained in a sterile cell culture environment and routinely tested for mycoplasma contamination. Irradiation was performed with a Faxitron Cabinet X-ray System Model RX-650. A-196 (Sigma-Aldrich) was applied for 72 h at a concentration of 1 μM. Transfections with Ambion Silencer Select siRNAs were performed for 72 h using Lipofectamine RNAiMAX (Thermo Fisher Scientific). The following Silencer Select siRNAs were used at a final siRNA concentration of 25 nM: siBRCA1 (s459) and siBARD1 (s1887). Negative Silencer Select control Neg1 from Ambion was used as non-targeting control. For pulsed EdU (5-ethynyl-2′-desoxyuridine) (Thermo Fisher Scientific) incorporation, cells were incubated for 20 min in a medium containing 10 μM EdU. The Click-iT EdU Alexa Fluor Imaging Kit (Thermo Fisher Scientific) was used for EdU detection.

### Immunofluorescence

Cells were grown on sterile 12 mm glass coverslips, fixed in 3% formaldehyde in PBS for 15 min at room temperature, washed once in PBS, permeabilized for 5 min at room temperature in PBS supplemented with 0.2% Triton X-100 (Sigma-Aldrich), and washed twice in PBS. All primary and secondary antibodies (Alexa fluorophores; Life Technologies) were diluted in filtered DMEM containing 10% FBS and 0.02% Sodium Azide. Antibody incubations were performed for 2 h at room temperature. After antibody incubations, coverslips were washed once with PBS and incubated for 10 min with PBS containing 4′,6-diamidino-2-phenylindole dihydrochloride (DAPI, 0.5 μg/ml) at room temperature to stain DNA. After three washing steps in PBS, coverslips were briefly washed with distilled water and mounted on 6 μl Mowiol-based mounting media. The following primary antibodies were used for immunostaining: H2AX Phospho S139 (mouse, 613401, 1:1,000; BioLegend),

53BP1 (mouse, Upstate MAB3802, 1:1,000), H4K20me2 (rabbit, ab9052, 1:100; Abcam), H4K20me1 (rabbit, ab9051, 1:200; Abcam), BRCA1 (mouse, sc-6954, 1:100; Santa Cruz), Cyclin A (mouse, sc-271682, 1:100; Santa Cruz), and RAD51 (rabbit, 70-002, 1:1,000; Bioacademia).

### QIBC

Automated multichannel wide-field microscopy for QIBC was performed on an Olympus ScanR Screening System equipped with an inverted motorized Olympus IX83 microscope, a motorized stage, IR-laser hardware autofocus, a fast emission filter wheel with single band emission filters, and a digital monochrome Hamamatsu ORCA-FLASH 4.0 V2 sCMOS camera (2,048 × 2,048 pixel, 12-bit dynamics) as described previously (Teloni et al, 2019). For each condition, image information of large cohorts of cells (typically at least 800 cells for the UPLSAPO 40× objective [NA 0.9], and at least 2000 cells for the UPLSAPO 20× objective [NA 0.75]) was acquired under non-saturating conditions at a single autofocus-directed z-position. Identical settings were applied to all samples within one experiment. Images were analyzed with the inbuilt Olympus ScanR Image Analysis Software Version 3.0.0, a dynamic background correction was applied, and detection of cell nuclei was performed using an integrated intensity-based object detection module based on the DAPI signal. All downstream analyses were focused on properly detected interphase nuclei containing a 2N–4N DNA content as measured by total and mean DAPI intensities. Fluorescence intensities were quantified and are depicted as arbitrary units. For normalization according to DNA content, measurement parameters (e.g., 53BP1 foci numbers) were divided at the single cell level by the DNA content (measured as total DAPI intensity per nucleus). The parameter resulting from this normalization has arbitrary units and was multiplied by a multiple of 10 to yield data that could be plotted on linear scale in the depicted range. Scatter plots of asynchronous cell populations were generated with Spotfire data visualization software (TIBCO). Within one experiment, similar cell numbers were

compared for the different conditions. Representative scatter plots and quantifications of independent experiments, typically containing several thousand cells each, are shown.

## Supplementary Information

## Acknowledgements

We are grateful to the University of Zurich Center for Microscopy and Image Analysis for microscopy support. We thank all members of our lab and of the Department of Molecular Mechanisms of Disease (DMMD) and of the Institute of Molecular Cancer Research (IMCR) for discussions, M Pruschy for input and advice on normalization of radiation damage, and A Groth for helpful comments on the manuscript. Research in the lab of M Altmeyer is supported by the Swiss National Science Foundation (grants 150690 and 179057), the European Research Council (ERC) under the European Union's Horizon 2020 research and innovation program (ERC-2016-STG 714326) and the University of Zurich Candoc & Postdoc program. The authors declare no competing interests.

### Author Contributions

J Michelena: data curation, formal analysis, investigation, visualization, methodology, and writing—review and editing.
S Pellegrino: data curation, formal analysis, investigation, visualization, methodology, and writing—review and editing.
V Spegg: data curation, formal analysis, investigation, visualization, methodology, and writing—review and editing.
M Altmeyer: conceptualization, resources, formal analysis, supervision, funding acquisition, visualization, writing—original draft, and project administration.

### Conflict of Interest Statement

The authors declare that they have no conflict of interest.

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
