## [Reviewer comments · Life Science Alliance]

Life Science Alliance

Replicated chromatin curtails 53BP1 recruitment in BRCA1-proficient and -deficient cells

Jone Michelena, Stefania Pellegrino, Vincent Spegg, and Matthias Altmeyer

DOI: <https://doi.org/10.26508/lsa.202101023>

Corresponding author(s): Matthias Altmeyer, Zurich, University

Review Timeline:	Submission Date:	2021-01-18
	Editorial Decision:	2021-03-15
	Revision Received:	2021-03-22
	Accepted:	2021-03-22

Scientific Editor: Shachi Bhatt

Transaction Report:

March 15, 2021

RE: Life Science Alliance Manuscript #LSA-2021-01023-T

Dr. Matthias Altmeyer
Zurich, University
Institute of Veterinary Biochemistry &
Molecular Biology
Zurich, Winterthurerstrasse 190 8057
Switzerland

Dear Dr. Altmeyer,

Thank you for submitting your revised manuscript entitled "Replicated chromatin curtails 53BP1 recruitment in BRCA1-proficient and -deficient cells". We apologize for the unusual and extended delay in getting back to you. We would be happy to publish your paper in Life Science Alliance pending revisions to meet all of the requests made by the reviewers and necessary to meet our formatting guidelines.

As you will note from the reviewers' comments below, the reviewers are quite enthusiastic about these findings and have asked for only minor revisions. We, thus, encourage you to submit a revised version back to us that addresses all of the reviewers' points.

Along with the points listed below, please also attend to the following,

- please consult our manuscript preparation guidelines <https://www.life-science-alliance.org/manuscript-prep> and make sure your manuscript sections are in the correct order;
- please separate the Results and Discussion section into two - 1. Results 2. Discussion, as per our formatting requirements
- please add ORCID ID for the corresponding author-you should have received instructions on how to do so
- please add a Category, Running Title, and Summary blurb/Alternate Abstract for your manuscript in our system
- please add Author Contributions of all Authors in our system
- please add a conflict of interest statement to your main manuscript text
- please upload your main and supplementary figures as single files
- please add your main and supplementary figure legends to the main manuscript text after the references section
- please add callouts for Figure 7A-C to your main manuscript text;
- LSA allows supplementary figures, but no EV Figures; please update your callouts for the Supplementary Figures in the manuscript Extended Data Figure 1a=Fig S1A; while supplementary figures use the system supplementary Fig S1;
- please make sure the manuscript sections are aligned in accordance with LSA's formatting guidelines: please separate the Figure legends and Supplemental Figure legends into separate sections;
- please change figure panels in the legends from lower case to Capital letters, i.e. from a) to A, etc., please change the callouts in the manuscript text as well

- please upload your main manuscript text as an editable doc file
- please add scale bars for Figures 1A, B and C

A. FINAL FILES:

B. MANUSCRIPT ORGANIZATION AND FORMATTING:

****The license to publish form must be signed before your manuscript can be sent to production. A link to the electronic license to publish form will be sent to the corresponding author only. Please**

take a moment to check your funder requirements.**

Sincerely,

Shachi Bhatt, Ph.D.

Executive Editor

Life Science Alliance

<https://www.lsajournal.org/>

Interested in an editorial career? EMBO Solutions is hiring a Scientific Editor to join the international Life Science Alliance team. Find out more here -

https://www.embo.org/documents/jobs/Vacancy_Notice_Scientific_editor_LSA.pdf

Reviewer #1 (Comments to the Authors (Required)):

Michelena et al present an elegant quantitative fluorescence microscopy setup to score 53BP1 foci in a cell cycle-resolved manner (using DNA content and cell cycle markers). Using this system they suggest that, in general, appropriate normalisation procedures should be more widely considered. Specifically, they show that 53BP1 recruitment to radiation-damaged replicated DNA is severely suppressed (compared to unreplicated DNA) in BRCA-proficient and moderately suppressed in BRCA-deficient cells. They suggest that replication-coupled dilution of H4K20me2 may act in itself to reduce 53BP1 recruitment, in addition to the well-established antagonism between 53BP1-RIF1-Shieldin and BRCA1-BARD1.

The manuscript is very well written and the data are nicely presented. However, the findings are not exactly novel (except for the speculation about the additional role of H4K20me2 dilution, which is far from conclusive). Yet, the authors have demonstrated an elegant methodological approach that may prove valuable to the community.

One point that should be briefly discussed is the question how alternative end joining (which seems to be quite common in tumour cells, whenever repair pathway choice mechanisms are disturbed) fits into this concept.

Reviewer #2 (Comments to the Authors (Required)):

In this manuscript the authors have utilized a quantitative image-based cytometry (QIBC)

technique to examine cellular stress response at single cell level. QIBC utilizes automated high-content microscopy in a cytometry-like fashion that allows direct assignment of cell cycle distribution of protein association to chromatin bypassing the requirement to synchronize the cell culture. It is well known that BRCA1 and 53BP1 play an important and antagonistic role in the pathway choice for the repair of double strand breaks by HR or NHEJ. The authors have used QIBC to examine how cells switch from a 53BP1-dominated to a BRCA1-dominated response during cell cycle. By normalizing for increased genome size in S and G2 phases as cells replicate their DNA, the authors have shown that 53BP1 recruitment to damaged replicated chromatin is inefficient in both BRCA1-proficient and BRCA1-deficient cells compared to its accumulation on the chromatin that has not yet replicated. The authors also showed the impact of dilution of H4K20me2 on the replicated chromatin during S and G2 phase may also play an important role in the binding of 53BP1 to the DSB sites.

Overall the findings are interesting and demonstrate the power of QIBC-based analysis. It allows single cell normalization to control for increase in genome size when the DNA undergoes replication. It can have a significant impact on data interpretation.

There are no major concerns. The only minor suggestion is to re-write a number of run-on sentences throughout the text. These long sentences make it extremely difficult to understand the message that is being conveyed. Just a few examples: Page 3, 1st paragraph 2nd sentence, 2nd paragraph, all three sentences. Page 5, 1st paragraph last sentence, 2nd paragraph, 4th sentence.

Reviewer #3 (Comments to the Authors (Required)):

Michelena and Altmeyer use QIBC, fluorescence microscopy and single cell analysis to examine 53BP1 foci intensity as a function of cell cycle and BRCA1/BARD1 expression. They demonstrate that 53BP1 has reduced association with replicating chromatin irrespective of BRCA1 status, albeit 53BP1 levels are increased following BRCA1/BARD1 knockdown in G2. The work agrees with elegant models put forth by the Groth group, showing that ankyrin repeats of BARD1 bind to H4K20me0 present on nascent nucleosomes in replicating DNA. The multiparameter, quantitative analyses and descriptions provide a nice contribution to the field and may be widely used in high content analysis screening approaches.

There are a few issues related to putting the findings in the context of other literature. This is important because the authors only perform correlative analysis and other possible explanations for their findings cannot be ruled out. In addition, the work does not adequately address other cell cycle studies that examine competition between BRCA1 and 53BP1 for DSB localization. These issues can be largely addressed in the text

Specific concerns

1. The authors should check protein expression levels of 53BP1 at different cell cycle phases to make sure this does not affect the results observed.
2. The authors have not yet offered enough direct evidence to definitively assign the diminished 53BP1 foci at replicated chromatin with the reduced H4K20me2 in late S/early G2. Since unreplicated chromatin (2N stage) has many differences compared to replicated chromatin (4N stage), this issue should be acknowledged as merely correlative.
3. Escribano-Díaz and Durocher Mol Cell 2013 PMID 23333306 demonstrate that BRCA1 foci substantially increase in G1 in Rif1/53BP1 deficient cells. This increase presumably occurs in the face of >80% of nucleosomes containing H4K20me2. Other papers have also shown competition between BRCA1 and 53BP1 foci formation in different cell cycle phases. Finally, 53BP1-RIF1-

Shieldin is clearly active in S/G2 to confer radial chromosome formation in the absence of BRCA1 or even in BRCA2 mutant cells that contain BRCA1. These points may be discussed in light of the data in Michelena et al.

Point-by-point response

Reviewer #1:

Michelena et al present an elegant quantitative fluorescence microscopy setup to score 53BP1 foci in a cell cycle-resolved manner (using DNA content and cell cycle markers). Using this system they suggest that, in general, appropriate normalisation procedures should be more widely considered. Specifically, they show that 53BP1 recruitment to radiation-damaged replicated DNA is severely suppressed (compared to unreplicated DNA) in BRCA-proficient and moderately suppressed in BRCA-deficient cells. They suggest that replication-coupled dilution of H4K20me2 may act in itself to reduce 53BP1 recruitment, in addition to the well-established antagonism between 53BP1-RIF1-Shieldin and BRCA1-BARD1.

The manuscript is very well written and the data are nicely presented. However, the findings are not exactly novel (except for the speculation about the additional role of H4K20me2 dilution, which is far from conclusive). Yet, the authors have demonstrated an elegant methodological approach that may prove valuable to the community.

One point that should be briefly discussed is the question how alternative end joining (which seems to be quite common in tumour cells, whenever repair pathway choice mechanisms are disturbed) fits into this concept.

We would like to thank the reviewer for having taken the time to read and evaluate our study, and for their positive assessment of our work. There is indeed evidence that 53BP1-Shieldin, by restraining DNA end resection, not only shifts the balance towards classical NHEJ, but also favors alternative end joining over HR. We have included this aspect in the discussion of our manuscript (page 9).

Reviewer #2:

In this manuscript the authors have utilized a quantitative image-based cytometry (QIBC) technique to examine cellular stress response at single cell level. QIBC utilizes automated high-content microscopy in a cytometry-like fashion that allows direct assignment of cell cycle distribution of protein association to chromatin bypassing the requirement to synchronize the cell culture. It is well known that BRCA1 and 53BP1 play an important and antagonistic role in the pathway choice for the repair of double strand breaks by HR or NHEJ. The authors have used QIBC to examine how cells switch from a 53BP1-dominated to a BRCA1-dominated response during cell cycle. By normalizing for increased genome size in S and G2 phases as cells replicate their DNA, the authors have shown that 53BP1 recruitment to damaged replicated chromatin is inefficient in both BRCA1-proficient and BRCA1-deficient cells compared to its accumulation on the chromatin that has not yet replicated. The authors also showed the impact of dilution of H4K20me2 on the replicated

chromatin during S and G2 phase may also play an important role in the binding of 53BP1 to the DSB sites.

Overall the findings are interesting and demonstrate the power of QIBC-based analysis. It allows single cell normalization to control for increase in genome size when the DNA undergoes replication. It can have a significant impact on data interpretation.

There are no major concerns. The only minor suggestion is to re-write a number of run-on sentences throughout the text. These long sentences make it extremely difficult to understand the message that is being conveyed. Just a few examples: Page 3, 1st paragraph 2nd sentence, 2nd paragraph, all three sentences. Page 5, 1st paragraph last sentence, 2nd paragraph, 4th sentence.

We would like to thank also this reviewer for their interest and endorsement. We have simplified and shortened the run-on sentences as suggested.

Reviewer #3:

Michelena and Altmeyer use QIBC, fluorescence microscopy and single cell analysis to examine 53BP1 foci intensity as a function of cell cycle and BRCA1/BARD1 expression. They demonstrate that 53BP1 has reduced association with replicating chromatin irrespective of BRCA1 status, albeit 53BP1 levels are increased following BRCA1/BARD1 knockdown in G2. The work agrees with elegant models put forth by the Groth group, showing that ankyrin repeats of BARD1 bind to H4K20me0 present on nascent nucleosomes in replicating DNA. The multiparameter, quantitative analyses and descriptions provide a nice contribution to the field and may be widely used in high content analysis screening approaches.

There are a few issues related to putting the findings in the context of other literature. This is important because the authors only perform correlative analysis and other possible explanations for their findings cannot be ruled out. In addition, the work does not adequately address other cell cycle studies that examine competition between BRCA1 and 53BP1 for DSB localization. These issues can be largely addressed in the text

We are grateful also to this reviewer for having taken the time to review our study and for insightful suggestions. Inspired by the comments of the reviewer we have included additional experiments with the SUV4-20 inhibitor A-196 to block H4K20me2 and thereby more causally address the relationship between this histone mark and 53BP1 recruitment, both in BRCA1-proficient and -deficient cells. The results are consistent with our model and show impaired 53BP1 recruitment when H4K20me2 is reduced, even in BRCA1-depleted cells (Figure S3 and S7). As H4K20me2 is lowered in these experiments but not completely lost, we cannot formally exclude that additional replication status-dependent chromatin features may also affect 53BP1 recruitment. We therefore discuss this possibility in the manuscript text (page 8 and 9).

Specific concerns

1. The authors should check protein expression levels of 53BP1 at different cell cycle phases to make sure this does not affect the results observed.

53BP1 levels are stable across different cell cycle phases as determined by QIBC. We have added this information (Figure S2E).

2. The authors have not yet offered enough direct evidence to definitively assign the diminished 53BP1 foci at replicated chromatin with the reduced H4K20me2 in late S/early G2. Since unreplicated chromatin (2N stage) has many differences compared to replicated chromatin (4N stage), this issue should be acknowledged as merely correlative.

The new results with A-196 to inhibit H4K20me2 formation (Figure S3 and S7) strengthen the link between this histone mark and 53BP1 recruitment. However, we agree that additional differences between unreplicated and replicated chromatin can play a role and have acknowledged this accordingly (page 8 and 9).

3. Escribano-Díaz and Durocher Mol Cell 2013 PMID 23333306 demonstrate that BRCA1 foci substantially increase in G1 in Rif1/53BP1 deficient cells. This increase presumably occurs in the face of >80% of nucleosomes containing H4K20me2. Other papers have also shown competition between BRCA1 and 53BP1 foci formation in different cell cycle phases. Finally, 53BP1-RIF1-Shieldin is clearly active in S/G2 to confer radial chromosome formation in the absence of BRCA1 or even in BRCA2 mutant cells that contain BRCA1. These points may be discussed in light of the data in Michelena et al.

We agree that the antagonism between 53BP1 and BRCA1 goes in both directions and have emphasized this, also referring to the studies by Escribano-Díaz et al., and by others in the discussion of the manuscript (page 9). The radial chromosome formation by 53BP1 in the absence of BRCA is very interesting and consistent with our results showing re-establishment of H4K20me2 in G2 with re-gained 53BP1 recruitment. We discuss this important point explicitly in the revised manuscript (page 10).

March 22, 2021

RE: Life Science Alliance Manuscript #LSA-2021-01023-TR

Dr. Matthias Altmeyer
Zurich, University
Institute of Veterinary Biochemistry &
Molecular Biology
Zurich, Winterthurerstrasse 190 8057
Switzerland

Dear Dr. Altmeyer,

Thank you for submitting your Research Article entitled "Replicated chromatin curtails 53BP1 recruitment in BRCA1-proficient and -deficient cells". It is a pleasure to let you know that your manuscript is now accepted for publication in Life Science Alliance. Congratulations on this interesting work.

DISTRIBUTION OF MATERIALS:

Again, congratulations on a very nice paper. I hope you found the review process to be constructive and are pleased with how the manuscript was handled editorially. We look forward to future exciting submissions from your lab.

Sincerely,

Shachi Bhatt, Ph.D.

Executive Editor

Life Science Alliance

<https://www.lsjournal.org/>

Interested in an editorial career? EMBO Solutions is hiring a Scientific Editor to join the international Life Science Alliance team. Find out more here -

https://www.embo.org/documents/jobs/Vacancy_Notice_Scientific_editor_LSA.pdf